# Edematous Dermal Thickening on Magnetic Resonance Imaging as a Biomarker for Lymphatic Surgical Outcomes

**DOI:** 10.3390/medicina59081369

**Published:** 2023-07-26

**Authors:** JacqueLyn R. Kinney, Sara Babapour, Erin Kim, Rosie Friedman, Dhruv Singhal, Bernard T. Lee, Leo L. Tsai

**Affiliations:** 1Division of Plastic and Reconstructive Surgery, Department of Surgery, Beth Israel Deaconess Medical Center, Harvard Medical School, Boston, MA 02215, USA; jacquelyn.kinney@rutgers.edu (J.R.K.);; 2Department of Radiology, Beth Israel Deaconess Medical Center, Harvard Medical School, Boston, MA 02215, USA

**Keywords:** breast cancer-related lymphedema, debulking, MRI, dermal thickness, lymphedema

## Abstract

*Background and Objectives*: One of the surgical treatments for breast cancer-related lymphedema (BCRL) is debulking lipectomy. The aim of this study is to investigate whether dermal thickness could be utilized as an objective indicator of post-operative changes following debulking. *Materials and Methods*: A retrospective review of BCRL patients who underwent debulking lipectomy was conducted. MRI-based dermal thickness was measured by two separate trained readers at 16 regions of the upper extremity. Pre- and post-operative reduction in dermal thickness was compared across the affected and unaffected (control) arms for each patient. The Wilcoxon rank sum test was used to assess for significant change. Univariate linear regression was used to assess the relationship between dermal thickness reduction and changes to LYMPH-Q scores, L-Dex scores, and relative volume change. *Results*: Seventeen patients were included in our analysis. There was significant reduction in dermal thickness at 5/16 regions in the affected arm. Dermal thickness change was significantly correlated with LYMPH-Q scores, L-Dex scores, and relative volume change in 2/16 limb compartments. There was predominant dermal thickening in the dorsal compartment of the upper arm and in the ventral and ulnar compartments of the forearm. *Conclusions*: Dermal thickness shows promising utility in tracking post-operative debulking procedures for breast cancer-related lymphedema. Further studies with larger patient populations and a variety of imaging modalities are required to continue to develop a clinically objective and reproducible method of post-surgical lymphedema staging and monitoring.

## 1. Introduction

Lymphedema is a chronic disease attributed to lymph accumulation in soft tissues, which results in reactive inflammation and swelling of the affected areas. This occurs from disruption of lymphatic flow, most commonly from oncologic therapy in the US [1]. Lymphedema can affect up to 20% of breast cancer patients [2], 37% of patients treated for gynecologic malignancies [3], and upwards of 90% of patients with head and neck cancers [4]. Current treatments for lymphedema include preventative immediate lymphatic reconstruction (ILR) [5], manual lymph drainage (MLD), compression, exercise, skincare, decongestive lymphedema therapy (DLT) [6], debulking lipectomy, vascularized lymph node transfer (VLNT), and lymphaticovenous anastomosis (LVA) [6].

Debulking lipectomy has historically been a non-homogenous form of lymphedema treatment, in that many surgeons have differing methodologies to complete the procedure. Such techniques include the use of liposuction or the direct surgical excision of subcutaneous tissue [7]. Our institution utilizes one of these such techniques, known as the Brorson technique, which utilizes liposuction to reduce the fat burden in patients with lymphedema [8]. The use of power-assisted liposuction and subsequent compression in those with chronic extremity lymphedema has been shown to not only decrease limb volumes but also reduce infection rates and improve patients’ quality of life [8]. Although debulking lipectomy can result in improved patient outcomes, many patient factors determine what the best treatment will be for their specific disease severity and tissue composition.

With the recent greater availability of surgical options for lymphedema, accurate pre-operative and post-treatment staging of disease is needed. Previous studies have demonstrated that even within the same lymphedema severity grading system, there is a wide variety of fat vs. fluid dominance and heterogeneity of appearance within the tissue [9]. At our institution, each patient’s individual fat vs. fluid dominance, as seen in pre-operative MRI imaging, is utilized in order to most effectively tailor treatment plans [10,11]. For patients with fat dominance, debulking is the preferred surgical treatment, followed by VLNT in one or two years [11]. For patients with fluid dominance, VLNT without an initial debulking has been our preferred method of initial surgical treatment [11]. The timing of surgery is highly dependent on how optimized the patients are either pre-surgery or after initial debulking when combined with VLNT. Clinical status is normally assessed using subjective patient questionnaires (LYMPH-Q), bioimpedance scores (L-Dex), and relative volume change in the limb of interest. However, these scores often do not correlate well with each other.

Quantitative imaging biomarkers have the potential to provide more consistent, objective measures of overall lymphatic status. We hypothesized, based on initial observations during multidisciplinary conferences, that dermal thickness, easily measurable on MRI scans, is positively correlated to the severity of subjacent lymphatic congestion and could therefore be used as an objective measure of post-surgical effectiveness. The aim of this study is to measure dermal thickness across upper extremities affected by breast cancer-related lymphedema (BCRL) to determine if regional or global changes in dermal thickness correlate with clinical improvement following debulking lipectomy.

## 2. Materials and Methods

A retrospective review was conducted at the Boston Lymphatic Center/Beth Israel Deaconess Medical Center. Institutional review board approval was obtained for this study (Protocol #2022P000670). A review of a prospectively maintained REDCap Quality Improvement Database [12] and supplemental chart review were performed. Debulking lipectomy in this study was defined as the use of power-assisted liposuction as a symptomatic treatment for those with chronic lymphedema. Patients 18 years or older who received a debulking lipectomy procedure for upper extremity BCRL from January 2015 to August 2022 were included. Patients were excluded if they were less than 18 years old, had debulking treatments for reasons other than lymphedema, had any surgery on either upper extremity, or had any prior history or clinical symptoms of lymphatic dysfunction such as primary lymphedema. Patient demographics, lymphedema characteristics, intraoperative variables, and surveillance data were extracted for analysis. Baseline characteristics were summarized using median and interquartile range (IQR) for continuous data and counts and percentages for categorial data.

At our institution, a formal diagnosis of lymphedema was defined as the presence of symptoms consistent with lymphedema, such as pain, swelling, restricted range of motion, trouble fitting into clothing, feelings of heaviness or tightness, and one of the following objective findings: (1) relative limb volume change greater than 10% or (2) Lymphedema Index (L-Dex) greater than 10 from baseline. LYMPH-Q is a subjective patient survey measuring patient-reported outcomes, with higher scores representing lower quality of life in patients suffering from lymphedema.

The dates of pre-operative and post-operative MRI were recorded. All upper extremity MRIs were included in the analysis. MRI was performed on a wide-bore 1.5 T magnet (Siemens Magneton Aera, Erlangen, Germany) using two 13-channel body array coils. Patients were instructed to remove compression garments for 48 h prior to the scan. Patients were positioned supine in a neutral position with their arms at the sides and palms facing medially. The target limb was positioned as close to the magnet isocenter as possible. A total of four imaging stations were acquired: non-affected upper arm (shoulder to elbow), unaffected forearm, affected upper arm, and affected forearm. At each of these stations, an axial T2-weighted fat-suppressed axial image was acquired with the following parameters: STIR (short-tau inversion recovery) sequence, for the upper arm, TR = 7080 ms, TE 53 ms, echo train length 16, field of view 400 × 200 mm, 52 slices, slice thickness 6 mm, matrix size 384 × 192; for the forearm, TR 7150 ms, TE 53 ms, echo train length 16, field of view 160 × 145 mm, 52 slices, slice thickness 6 mm, matrix size 192 × 174. Phase encoding was performed in the anterior–posterior direction.

Dermal thickness was measured by two separate readers (J.K. and S.B.) at 4 points (medial/ulnar, lateral/radial, posterior/dorsal, and anterior/ventral), at 2 different locations in the upper arm (UA1 and UA2) and 2 different locations in the forearm (F1 and F2), totaling 16 sites per arm (Figure 1). For each patient, the unaffected arm was set as the control for the affected arm, which was not altered by the debulking procedure or prior surgeries, and therefore it was hypothesized to have no substantial change in dermal thickness post-operatively. Maximum dermal thickness was designated to be the anatomical location with the greatest dermal thickness measurement. Data were compared across the debulking and control arms.

### Statistical Analysis

An intraclass correlation coefficient (ICC) two-way random-effects model was used to rate interrater reliability. A Wilcoxon rank sum test was used to compare dermal thickness changes between the debulking and control groups. Univariate linear regression assessed the relationship between changes in dermal thickness and changes in L-Dex score, LYMPH-Q score, and limb volume change. All analyses were conducted for the data obtained by reader 1 (J.K.) and reader 2 (S.B.) separately.

A value of *p* < 0.05 was considered statistically significant. All analyses were performed using R statistical software (version 4.2.1, Vienna, Austria).

## 3. Results

Seventeen patients met the inclusion criteria and were included in this study (17 arms that underwent debulking lipectomy and 17 control arms). The median age was 67 (IQR 60–71), and 100% of patients were female. The population demographics can be seen in Table 1. The ICC between readers was 0.05, demonstrating moderate reliability.

Both readers reached consensus in identifying the ulnar and ventral aspects of the forearm and the dorsal aspects of the upper arm to the areas of maximal dermal thickness. A visual distribution of these locations can be seen in Figure 2.

There was significant dermal thickness reduction following debulking, using the contralateral side as reference, in 8/16 regions identified by reader 1 (Table 2) and 8/16 regions identified by reader 2 (Table 3). Readers 1 and 2 reached a consensus in identifying significant reduction in 5/16 regions: F1 radial (*p* < 0.001, *p* = 0.010; reader 1 and 2 *p*-values, respectively), F2 radial (*p* = 0.005, *p* = 0.004), UA1 ventral (*p* = 0.039, *p* = 0.016), UA2 dorsal (*p* = 0.014, *p* = 0.025), and UA2 lateral (*p* = 0.002, *p* = 0.034). For all these regions, the median change was −1 mm. The median change across the control arms was 0 mm across all regions.

Debulking was associated with an improvement in L-Dex score (median −15.1, IQR: −33.3 to −3.7), LYMPH-Q score (median −12.0, IQR: −18.5 to −11.5), and percent volume difference (median −30.9, IQR: −41.0 to −20.7) after the debulking surgery. Table 4 and Table 5 display correlations between these scores and changes to dermal thicknesses, according to measurements interpreted by reader 1 and 2, respectively. Across both readers, there were significant correlations with changes to LYMPH-Q, particularly in the F1 ventral region (*p* = 0.021, *p* = 0.004, *p*-values for reader 1 and 2, respectively). In addition, changes to the UA1 dorsal region also correlated significantly with changes to relative volume (*p* = 0.046, *p* = 0.015).

## 4. Discussion

In this study, we investigated the utility of dermal thickness on MRI as a biomarker for post-operative assessment of patients who received debulking lipectomy for BCRL. These results demonstrated a significant decrease in dermal thickness in 5/16 regions of the debulking arm compared to the control arm, and an association of these changes to LYMPH-Q scores in the ventral and ulnar regions of the forearm. At this institution, MRIs are already used to track other changes in lymphedema such as the volume of fat and subcutaneous fluid, and the dermis is always included in the field of view. The measurement of dermal thickening can be carried out quickly on any standard clinical workstation without additional sequences, contrast, or post-processing. In places where MRI may not be as readily available, dermal thickening may be measured with other techniques, such as ultrasound, offering potentially faster and cheaper ways to assess post-debulking patients.

Dermal thickening was an observation made during the development and validation of an MRI staging system for upper extremity lymphedema; this was thought to be attributed to lymphatic congestion in areas of dermal backflow [13,14]. Increasing fibrosis of surrounding anatomical structures in areas afflicted with lymphedema may also contribute to dermal thickening [15]. Fibrosis of the dermis in patients with chronic lymphedema has even been incorporated into what the International Society of Lymphology has used in their clinical staging criteria [16]. According to the staging determined by this society, Stage 1 includes lymphedema that improves with limb elevation. Stage 2 includes pitting edema that does not improve with limb elevation. Finally, Stage 3 represents skin changes and fibroadipose tissue deposition that can result in thickening of the skin and soft tissues surrounding the area of edema [16]. Although different staging systems and grading of lymphedema severity exist [17], late-stage disease is often accompanied by increased fibrosis, which can contribute to dermal thickening. In our study, we focused on edematous thickening, which can be seen in both non-fibrotic edema as well as fibrosis. Dermal thickness appears to be a reversible process in patients with chronic lymphedema, as prior studies have demonstrated that it is possible to significantly decrease dermal thickness [18]. Irreversible skin thickening would suggest scarring, which should result in a lack of signal intensity on the T2-weighed sequences, which was not observed in our cohort. Furthermore, our results are concordant with prior observations where debulking was associated with decreased dermal backflow and improved lymphatic function on lymphoscintigraphy [19].

Recent studies have shown that lymphedema does not affect the limb uniformly [15]. Friedman et al. demonstrated that lymphedema of the forearm predominantly distributed in the ulnar aspect followed by the volar aspect [20]. Within the upper arm, the majority of the fluid infiltration was seen distally and posteriorly [20]. This was corroborated by our results, as there was a similar distribution in maximal dermal thickening. This lends further support to the notion that regional dermal thickening reflects the severity of lymphatic congestion in that area. Notably, these areas of measured maximal dermal thickness significantly exceeded the control arm’s respective measurements, which correlated well with known estimations of non-diseased dermal thickness (1–4 mm) [21].

Even though L-Dex, LYMPH-Q, and volume difference improved with debulking, there was less of a direct association between these clinical measures and dermal thickness measurements. This is likely because the clinical measures reflect overall lymphedema presentation and symptoms, while there is a notable uneven distribution of dermal thickening across different regions. The ventral forearm and posterior upper arm were areas where there was some association between dermal thickness change and clinical measures, which reflects areas where lymphedema manifests at an earlier stage and where the most edema is typically seen in severe cases. The correlation between LYMPH-Q and dermal thickness at the ulnar and ventral forearm matches areas where there are typically the most symptoms. We expect that studies with a larger patient cohort and greater statistical power will reveal further associations.

### Limitations

There are several limitations to this study. First, all patients included in this study were female, a situation that does not completely represent all patients who may suffer from breast cancer-related lymphedema. Furthermore, the sample size was relatively small, precluding more detailed comparisons between regional dermal thickening and global clinical measures, as discussed above, but it was adequate to demonstrate clear reductions in dermal thickening following debulking. Given the trends in correlation between MRI and clinical measures, a larger study should be pursued to further validate the use of dermal thickening for clinical assessment of lymphedema status. We also limited our study to BCRL patients with unilateral lymphedema, and further studies in patients with other causes of lymphedema and those with lower extremity lymphedema are needed. Lastly, we did not track longer-term outcomes of debulking, which may have allowed for a comparison of patients with good versus suboptimal outcomes, and whether measurements of dermal thickness could be used to assess those differences.

## 5. Conclusions

Dermal thickness, as measured on MRI, decreases following debulking surgery for patients with breast cancer-related lymphedema, concordant with improvements in clinical measures, therefore offering a potential quantitative and regional biomarker that can be used to track surgical outcomes.

## Figures and Tables

**Figure 1 medicina-59-01369-f001:**
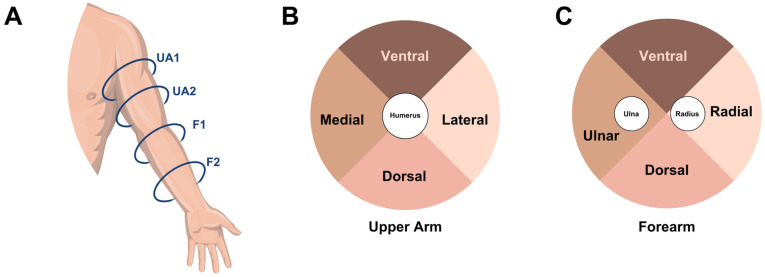
Anatomical location of dermal thickness measurements by quadrants. (**A**) Four anatomical locations of the MRI sections analyzed. UA1 is upper arm location 1, UA2 is upper arm location 2, F1 is forearm location 1, and F2 is forearm location 2. (**B**) Divisions of quadrants within each upper extremity MRI section analyzed. (**C**) Divisions of quadrants within each forearm MRI section analyzed.

**Figure 2 medicina-59-01369-f002:**
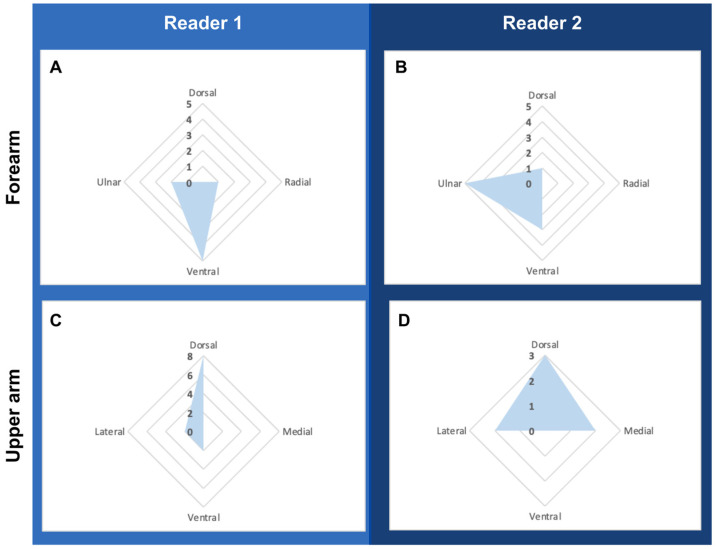
Maximal dermal thickness distribution by quadrant location, with the graph designating the number of patients at each location. (**A**) Reader 1’s relative distribution of overall maximal dermal thickness in all forearm measurements. (**B**) Reader 2’s relative distribution of overall maximal dermal thickness in all forearm measurements. (**C**) Reader 1’s relative distribution of overall maximal dermal thickness in all upper arm measurements. (**D**) Reader 2’s relative distribution of overall maximal dermal thickness in all forearm measurements.

**Table 1 medicina-59-01369-t001:** Characteristics of the total patient population.

Characteristics	N = 17
Age median (IQR)	67 (60–71)
Gender n (%)	
Female	17 (100%)
Race n (%)	
White	14 (82%)
Black	1 (6%)
Asian	1 (6%)
Unknown	1 (6%)
Ethnicity n (%)	
Non-Hispanic	17 (100%)
Hispanic	0 (0%)
Months Elapsed Between Debulking and Post-Operative MRI median (IQR)	13 (12–14)

IQR—interquartile range.

**Table 2 medicina-59-01369-t002:** Differences between post-operative and pre-operative dermal thickness measurements: Interpreted by Reader 1.

Location of Measurement	Debulking (Δ mm) N = 17	Control (Δ mm) N = 17	*p*-Value
F1 Dorsal	0 (−0.1, 0) ^1^	0 (−0.1, 0)	0.529
F1 Radial	−1 (−1.7, 0)	0 (−0.2, 0)	<0.001 **^2^
F1 Ulnar	−1 (−0.9, 0)	0 (−0.2, 0)	0.002 **
F1 Ventral	0 (−0.5, 0)	0 (−0.2, 0)	0.187
F2 Dorsal	0 (−1.2, 0)	0 (−0.1, 0)	0.085
F2 Radial	−1 (−1.7, 0)	0 (−0.2, 1)	0.005 **
F2 Ulnar	−1 (−0.9, 0)	0 (−0.4, 0)	0.054
F2 Ventral	0 (−0.6, 0)	0 (−0.2, 0)	0.752
UA1 Dorsal	−1 (−1.4, −1)	0 (−0.3, 0)	0.153
UA1 Lateral	−1 (−1.0, 0)	0 (−0.1, 0)	0.005 **
UA1 Medial	0 (−0.5, 0)	0 (−0.1, 0)	0.776
UA1 Ventral	0 (−1.4, 0)	0 (0.0, 0)	0.039 **
UA2 Dorsal	−1 (−2.6, 0)	0 (0.0, 0)	0.014 **
UA2 Lateral	−1 (−1.8, 0)	0 (0.0, 0)	0.002 **
UA2 Medial	−1 (−1.5, 0)	0 (0.0, 0)	0.065
UA2 Ventral	−1 (−1.4, 0)	0 (−0.1, 0)	0.009 **

^1^ Negative values denote reduction in post-operative dermal thickness. ^2^ ** = significance (*p* < 0.05).

**Table 3 medicina-59-01369-t003:** Differences between post-operative and pre-operative dermal thickness measurements: interpreted by Reader 2.

Location of Measurement	Debulking (Δ mm) N = 17	Control (Δ mm) N = 17	*p*-Value
F1 Dorsal	0 (−1.0, 0) ^1^	0 (−0.2, 0)	0.110
F1 Radial	−1 (−1.2, 0)	0 (−0.1, 1)	0.010 **^2^
F1 Ulnar	−1 (−1.6, 0)	0 (−0.4, 1)	0.210
F1 Ventral	0 (−1.0, 0)	0 (−0.1, 0)	0.030 **
F2 Dorsal	0 (−0.7, 0)	0 (−0.2, 0)	0.454
F2 Radial	−1 (−1.1, 0)	0 (−0.2, 0)	0.004 **
F2 Ulnar	−1 (−1.0, 0)	0 (−0.6, 0)	0.018 **
F2 Ventral	0 (−1.0, 0)	0 (−0.3, 0)	0.026 **
UA1 Dorsal	0 (−0.5, 0)	0 (0.0, 1)	0.470
UA1 Lateral	0 (−0.6, 0)	0 (−0.4, 0)	0.552
UA1 Medial	0 (−0.5, 0)	0 (0.0, 1)	0.448
UA1 Ventral	0 (−0.8, 0)	0 (−0.2, 0)	0.016 **
UA2 Dorsal	0 (−1.7, 0)	0 (−0.3, 0)	0.025 **
UA2 Lateral	0 (−1.0, 0)	0 (−0.3, 0)	0.034 **
UA2 Medial	0 (−1.0, 0)	0 (−0.4, 0)	0.391
UA2 Ventral	0 (−0.8, 0)	0 (−0.5, 0)	0.735

^1^ Negative values denote reduction in post-operative dermal thickness. ^2^ ** = significance (*p* < 0.05).

**Table 4 medicina-59-01369-t004:** Correlations between changes in dermal thickness and changes to other clinical markers: Reader 1.

Location of Measurement	L-Dex Scores	LYMPH-Q Scores	Relative Volume Change ^1^
F1 Dorsal	0.300	0.254	0.910
F1 Radial	0.240	0.244	0.641
F1 Ulnar	0.874	0.588	0.033 **^2^
F1 Ventral	0.643	0.021 **	0.200
F2 Dorsal	0.478	0.083	0.200
F2 Radial	0.587	0.425	0.572
F2 Ulnar	0.342	0.010 **	0.351
F2 Ventral	0.242	0.843	0.536
UA1 Dorsal	0.008 **	0.678	0.046 **
UA1 Lateral	0.367	0.299	0.860
UA1 Medial	0.273	0.788	0.577
UA1 Ventral	0.960	0.704	0.499
UA2 Dorsal	0.585	0.413	0.420
UA2 Lateral	0.795	0.435	0.869
UA2 Medial	0.481	0.375	0.498
UA2 Ventral	0.766	0.336	0.470

^1^ All values listed are *p*-values calculated via a linear regression model. ^2^ ** = significance (*p* < 0.05).

**Table 5 medicina-59-01369-t005:** Correlations between changes in dermal thickness and changes to other clinical markers: Reader 2.

Location of Measurement	L-Dex Scores	LYMPH-Q Scores	Relative Volume Change ^1^
F1 Dorsal	0.421	0.883	0.077
F1 Radial	0.168	0.556	0.497
F1 Ulnar	0.729	0.378	0.083
F1 Ventral	0.933	0.004 **^2^	0.299
F2 Dorsal	0.377	0.710	0.133
F2 Radial	0.287	0.895	0.992
F2 Ulnar	0.935	0.311	0.622
F2 Ventral	0.451	0.578	0.146
UA1 Dorsal	0.935	0.422	0.015 **
UA1 Lateral	0.659	0.348	0.881
UA1 Medial	0.878	0.364	0.107
UA1 Ventral	0.383	0.310	0.330
UA2 Dorsal	0.621	0.151	0.127
UA2 Lateral	0.602	0.198	0.987
UA2 Medial	0.052	0.054	0.478
UA2 Ventral	0.934	0.083	0.596

^1^ All values listed are *p*-values calculated via a linear regression model. ^2^ ** = significance (*p* < 0.05).

## Data Availability

The data are not publicly available due to privacy restrictions but can be provided by the authors on request.

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
