# Peer review of "Edematous Dermal Thickening on Magnetic Resonance Imaging as a Biomarker for Lymphatic Surgical Outcomes"

_medicina, 2023, doi:10.3390/medicina59081369_

Round 1
Reviewer 1 Report
This interesting work facing the role of dermal thikness in the evaluation of the lymphedema after debulking. I consider that this work can be relevant in the evaluation of lymphedema surgery and it is original. Nevertheless there are some points that must be improved.
Debulking techniques may be different. I think that authors must explain how debunking was done (liposuction, skin and subcutaneous tissue removal...).
It must be reflected in introduction and along the text.
Correlation with LYMPH-Q is amazing and it must be emphasized. Unfortunately the number of patients is very low.
I miss more information about the evolution time of the lymphedema itself, the period that passed after surgery and the type of lymphedema. On the other hand, all patients are female. You must conclude that it has been demostraste in women exclusively.
Author Response
Point 1. Debulking techniques may be different. I think that authors must explain how debunking was done (liposuction, skin and subcutaneous tissue removal...). It must be reflected in introduction and along the text.
Response to Point 1: Thank you for this feedback, we have added in additional background into the types of debulking that are preformed and as well as evidence to support the type of debulking that is performed at this institution. Furthermore, in the methods this is further outlined to clarify our inclusion criteria.
Point 2. Correlation with LYMPH-Q is amazing and it must be emphasized. Unfortunately the number of patients is very low.
Response to Point 2. Thank you, we have made a point to emphasize this correlation with LYMPH-Q at the end of our discussion. We also acknowledge the low patient population in our limitations section and further emphasized the need for a larger study to test additional trends we observed between MRI measures and other clinical measures.
Point 3. I miss more information about the evolution time of the lymphedema itself, the period that passed after surgery and the type of lymphedema. On the other hand, all patients are female. You must conclude that it has been demonstrate in women exclusively.
Response to Point 3. Thank you for this feedback. To our knowledge, there has not been a study outlining the timeline in which fibrosis decreases after debulking. As this is a pilot study using MRI to track dermal thickness, we believe this to be an important aspect of the disease process to keep in mind moving forward and will hopefully be able to provide a timeline on the regression of fibrosis/dermal thickness in future studies. We have however added in additional information in the discussion on the staging and progression of lymphedema to clarify when fibrosis is commonly seen in the stages of the disease. Additionally, the limitations section has been expanded to note the limitation this study presents due to its homogenously female cohort.
Reviewer 2 Report
First of all, I would like to thank you for inviting me to review the manuscript entitled: “Edematous Dermal Thickening on Magnetic Resonance Imaging as a Biomarker for Lymphatic Surgical Outcomes.’
The manuscript investigates the usefulness of MRI as a biomarker to assess dermal thickness post-operatively in patients who received debulking lipectomy for breast cancer-related lymphedema. The discussion section explains the results in the context of published information. The conclusions accurately and clearly explain the main clinical message. The figures and tables are of good quality. The references are appropriate and current.
I can find no major flaws in the manuscript. The only limitation is the small sample size. If similar results are presented in a larger study, then edematous dermal thickening assessed by MRI can be a useful biomarker for lymphatic surgical outcomes.
The manuscript is well written in terms of clarity, style, and use of English and has a logical construction.
Author Response
Point 1. First of all, I would like to thank you for inviting me to review the manuscript entitled: “Edematous Dermal Thickening on Magnetic Resonance Imaging as a Biomarker for Lymphatic Surgical Outcomes.’
The manuscript investigates the usefulness of MRI as a biomarker to assess dermal thickness post-operatively in patients who received debulking lipectomy for breast cancer-related lymphedema. The discussion section explains the results in the context of published information. The conclusions accurately and clearly explain the main clinical message. The figures and tables are of good quality. The references are appropriate and current.
I can find no major flaws in the manuscript. The only limitation is the small sample size. If similar results are presented in a larger study, then edematous dermal thickening assessed by MRI can be a useful biomarker for lymphatic surgical outcomes.
Response to Point 1. Thank you so much for you feedback. We did expand our discussion about sample size limitation as also suggested by reviewers 1 and 3.
Reviewer 3 Report
In the present manuscript, the authors demonstrated the use of dermal thickness as an indicator of postoperative changes after debulking. Although the study is novel and of immense interest in the field, I have a few minor concerns about the work as follows:
1. The sample size is very low. It will be good if the authors conduct their experiments on additional samples.
2. Is there any influence of debulking arm on the other arm (the authors used as control arm)? What is the dermal thickness of the healthy individual? The proper control could be the arm of a healthy individual. Please discuss.
3. Line 191, page 8: ...demonstrated the lymphedema of in the forearm.... Please remove one of the prepositions.
Author Response
In the present manuscript, the authors demonstrated the use of dermal thickness as an indicator of postoperative changes after debulking. Although the study is novel and of immense interest in the field, I have a few minor concerns about the work as follows:
Point 1. The sample size is very low. It will be good if the authors conduct their experiments on additional samples.
Response to Point 1. Thank you, we acknowledge this in our limitations section and highlight the need for a larger study to test additional trends we observed between MRI measures and other clinical measures.
Point 2. Is there any influence of debulking arm on the other arm (the authors used as control arm)? What is the dermal thickness of the healthy individual? The proper control could be the arm of a healthy individual. Please discuss.
Response to Point 2. Our data showed no change in the dermal thickness of the contralateral arm, which we have included in the results. We felt that the contralateral arm serves well as an internal control based on that finding since we were focusing on secondary lymphedema and none of our patients had lymphatic disease in the contralateral arm prior to surgery. We had also ensured that patients did not have prior surgeries on the contralateral arm or history of primary lymphedema, or other lymphatic disorders, prior to their breast-cancer related treatments, and we added this as exclusion criteria in the Methods section.
Point 3. Line 191, page 8: ...demonstrated the lymphedema of in the forearm.... Please remove one of the prepositions.
Response to Point 3. Thank you for noticing this error, this is now corrected.